# The evolution of acoustic size exaggeration in terrestrial mammals

Benjamin D. Charlton[1] & David Reby[2]

Recent studies have revealed that some mammals possess adaptations that enable them to produce vocal signals with much lower fundamental frequency ($F0$) and formant frequency spacing ($\Delta F$) than expected for their size. Although these adaptations are assumed to reflect selection pressures for males to lower frequency components and exaggerate body size in reproductive contexts, this hypothesis has not been tested across a broad range of species. Here we show that male terrestrial mammals produce vocal signals with lower $\Delta F$ (but not $F0$) than expected for their size in mating systems with greater sexual size dimorphism. We also reveal that males produce calls with higher than expected $F0$ and $\Delta F$ in species with increased sperm competition. This investigation confirms that sexual selection favours the use of $\Delta F$ as an acoustic size exaggerator and supports the notion of an evolutionary trade-off between pre-copulatory signalling displays and sperm production.

[1] School of Biology and Environmental Science, Science Centre West, University College Dublin (UCD), Belfield, Dublin 4, Ireland. [2] Mammal Vocal Communication and Cognition Research Group, School of Psychology, University of Sussex, Sussex BN1 9QH, UK. Correspondence and requests for materials should be addressed to B.C. (email: benjamin.charlton@ucd.ie).

dentifying the proximate and ultimate factors that underlie the extraordinary diversity of mammal vocal signals is a key objective of animal communication research, and an essential prerequisite for understanding the origins and evolution of human vocal communication[1]. Body size is known to exert major constraints on the frequency characteristic of animal vocalizations, and as a consequence, size differences between species explain a large proportion of the acoustic diversity of animal vocalizations[2,3]. Large animals tend to produce lower-pitched calls than smaller ones because they have larger larynges with longer vocal folds that can oscillate periodically at lower frequencies and longer vocal tracts that produce lower resonances (termed formant frequencies in animal vocalizations)[4–7]. While this general rule of acoustic allometry is broadly verified across mammal species, several exceptions, typically affecting male vocal signals, have been documented.

For example, some species possess anatomical innovations that enable males to produce abnormally low fundamental frequency (hereafter $F0$), such as the fleshy vocal pads of roaring cats[8], Mongolian and goitered gazelles[9,10] or saiga antelope[11]; hypertrophied larynges in howler and colobus monkeys[12], fallow deer[13] and hammer-headed bats[14], and even an additional, non-laryngeal set of vocal folds in the koala[15]. In other species, males produce abnormally low formant frequency spacing ($\Delta F$) for their size by extending their vocal tracts using descended and/or mobile larynges[10,13,16–19], additional resonators[20,21], or nasal proboscises[11,22,23]. Because these anatomical adaptations are often only present or disproportionately large in males and involved in the production of mating calls, it is generally assumed that they have evolved via selection pressures for individuals to lower frequency components to broadcast an exaggerated impression of their body size in reproductive contexts[1,17,24]. However, while this assumption has been verified experimentally within a small number of species[25–29], whether sexual selection pressures on male body size drive the evolution of putative acoustic size exaggeration across a wider range of mammalian taxa remains to be investigated.

Although phylogenetically controlled comparative analyses of vocal behaviour exist for birds[30,31], previous comparative investigations of mammal vocalizations are either restricted to one mammalian order (for example, Primates[3,32,33], Rodentia[34]) or family (for example, Cervidae[35], Felidae[36]). In this paper we provide the first phylogenetically controlled comparative examination of the selection pressures that lead to acoustic size exaggeration across nine orders and 72 species of terrestrial mammals. We show that the principle of acoustic allometry is generally observed across taxa, and that males from mating systems with strong selection pressures for large male body size produce lower $\Delta F$, but not $F0$, than expected for their size. Our findings also confirm that selection pressures to exaggerate size are relaxed in male species with larger testes relative to overall body size, indicating that a high level of post-copulatory sperm competition reduces the importance of pre-copulatory acoustic size exaggeration.

## Results

**Model selection criteria.** To test our hypotheses we used phylogenetic generalized least-squares (PGLS) regressions that simulated five different evolutionary scenarios. To select the best supported PGLS regression models, we started with a 'global' model including male body mass, habitat (arboreal or terrestrial), call-type (sexual or nonsexual), mating system (polygynous, monogamous, polyandrous, promiscuous or variable) and sexual size dimorphism or relative testes size depending on the hypothesis that was being tested, and iterated through all variable

combinations to explain variation in male $F0$ and $\Delta F$ for each of the five different evolutionary scenarios. All models considered included $\log_{10}$ male body mass to control for body size differences across species and a model selection criteria based on the Akaike's Information Criteria corrected for sample size (AICc) was used, in which the model having the lowest AICc value is chosen[37] (Supplementary Tables 1–6).

**Male body size versus $F0$ and $\Delta F$ across species.** Our model selection approach indicated that the best supported PGLS regression models to test for the effect of male body size on male $F0$ and $\Delta F$ were a Brownian motion model of evolution using Pagel's lambda ($\lambda$) to model the covariance structure (BM + $\lambda$) and a pure Brownian motion model (BM) with habitat included as a covariate, respectively (Supplementary Tables 1 and 2). The PGLS regressions showed that greater male body mass predicted lower $\log_{10} F0$ (estimate ± s.e. = − 0.50 ± 0.09, $\lambda = 0.87$, $t_{4,65} = − 5.92$, $P < 0.001$) and $\log_{10} \Delta F$ (estimate ± s.e. = − 0.34 ± 0.05, $t_{4,32} = − 6.19$, $P < 0.001$), confirming that the expected acoustic allometry exists across species (Fig. 1). We also found that arboreal species produced significantly lower $\log_{10} \Delta F$ than other terrestrial mammals (estimate ± s.e. = 0.30 ± 0.11, $t_{4,32} = 3.10$, $P = 0.008$).

**Male size dimorphism versus $F0$ and $\Delta F$ across species.** A BM + $\lambda$ model of evolution including $\log_{10}$ male body mass as a covariate best explained the relationship between size dimorphism and $F0$ (Supplementary Table 3). This model showed that size dimorphism was not significantly related to $\log_{10} F0$ (estimate ± s.e. = − 4.93 ± 3.11, $\lambda = 0.87$, $t_{5,64} = − 1.58$, $P = 0.119$) (Fig. 2a). The relationship between size dimorphism and $\Delta F$ was best explained by a pure Brownian motion model of evolution with $\log_{10}$ male body mass and habitat included as covariates (Supplementary Table 4). This model revealed that species with greater male sexual size dimorphism produced sexual calls with lower $\log_{10} \Delta F$ (estimate ± s.e. = − 3.58 ± 1.21, $t_{5,31} = − 2.97$, $P = 0.006$) (Fig. 2b, Supplementary Table 4), indicating that males produce vocal signals with lower than expected $\Delta F$ for their size in mating systems with sexual selection pressures for large male body size.

**Sperm competition versus $F0$ and $\Delta F$ across species.** The best supported models to examine the effect of post-copulatory sperm competition on $F0$ and $\Delta F$ were an Ornstein–Uhlenbeck (OU) model and a BM + $\lambda$ model of evolution, respectively (lowest AICc values, see Supplementary Tables 5 and 6). Both models included $\log_{10}$ male body mass as a covariate. The relationship between relative testes size and male acoustic values in the 42 mammal species for which acoustic and testes data were available, revealed that $\log_{10}$ relative testes size was positively correlated with $\log_{10} F0$ (estimate ± s.e. = 0.39 ± 0.16, $\alpha = 0.02$, $t_{5,39} = 2.50$, $P = 0.017$) (Fig. 3a). We also found that $\log_{10}$ relative testes size was positively correlated with $\log_{10} \Delta F$ (estimate ± s.e. = 0.09 ± 0.02, $\lambda = 1.02$, $t_{5,21} = 4.04$, $P < 0.001$) (Fig. 3b) for the 24 species with available acoustic and testes data. These findings indicate that species with larger testes relative to body size produce calls with higher $F0$ and $\Delta F$.

## Discussion

Several interesting results emerge from this phylogenetically controlled examination of the link between acoustic variation in mammal calls and putative pre- and post-copulatory sexual selection pressures in a wide range of mammalian species. First, the key predictions of the acoustic allometry are confirmed: males from larger species produce calls with lower $F0$ and lower

formants (Fig. 1). The analysis also reveals that males of arboreal species give sexual calls with lower $\Delta F$ than other terrestrial mammals. This finding is consistent with the notion that low frequency calls given from relatively higher positions are less affected by ground interference[38] and/or that lower frequencies propagate best in forest environments[39]. Interestingly, the relationship between body mass and $F0$ across nine orders of terrestrial mammals (Fig. 1a) indicates that laryngeal enlargement and the concomitant lengthening of the vocal folds is a more effective way of lowering $F0$ than thickening the vocal folds to increase their mass[40], and suggests that vocal pads may primarily support the production of high-amplitude low-$F0$ sexual calls, rather than lowering $F0$ *per se*.

A very close relationship between male $\Delta F$ and body mass is also revealed (Fig. 1b), illustrating how strong anatomical constraints affect the correlations between vocal tract length, skull size and overall body size[41]. When species without anatomical adaptations to lower formant frequencies are considered alone, the relationship is even stronger ($R^2$ of 0.79) (Fig. 1b). Interestingly, species with specific anatomical and/or behavioural adaptations that allow them to escape these constraints follow a separate downward shifted trend that still, nevertheless, represents a close relationship between $\Delta F$ and body mass ($R^2$ of 0.58) (Fig. 1b). This is suggestive of secondary constraints acting on size exaggerators, such as the sternum preventing any further laryngeal descent[16] and/or other skeletal structures that limit further enlargement of acoustic resonators,

which in turn limits the extent of size exaggeration and maintains a parallel allometric relationship between $\Delta F$ and body mass. Of particular interest are species that are not known to possess adaptations to lower $F0$ or formants yet still produce call frequencies which fall way below the expected acoustic allometry (for example, European badgers and mole rats). Future studies should further investigate these species' vocal anatomy in conjunction with the selection pressures acting on their vocal communication systems.

When investigating the effect of sexual selection for large male body size we found that sexual size dimorphism did not predict $F0$ across taxa. The lack of a relationship between sexual size dimorphism and $F0$ is not surprising, as $F0$ is generally a poor predictor of adult male body mass within species[5]. Our results, therefore, support the hypothesis that sexual selection does not systematically favour the use of $F0$ as an acoustic size exaggerator. In contrast, sexual size dimorphism was negatively correlated to formant frequency spacing, with greater male sexual size dimorphism resulting in male sexual calls with lower than expected $\Delta F$. This relationship indicates that sexual selection for increased male body size is likely to be a key force leading to the evolution of anatomical and/or behavioural adaptations that enable male callers to acoustically exaggerate their apparent body size via formant lowering. Although $\Delta F$ is known to function as a size exaggerator in some mammalian species[25–29], the findings of the current study constitute the first demonstration that sexual selection is a key driver of acoustic variability across mammals.

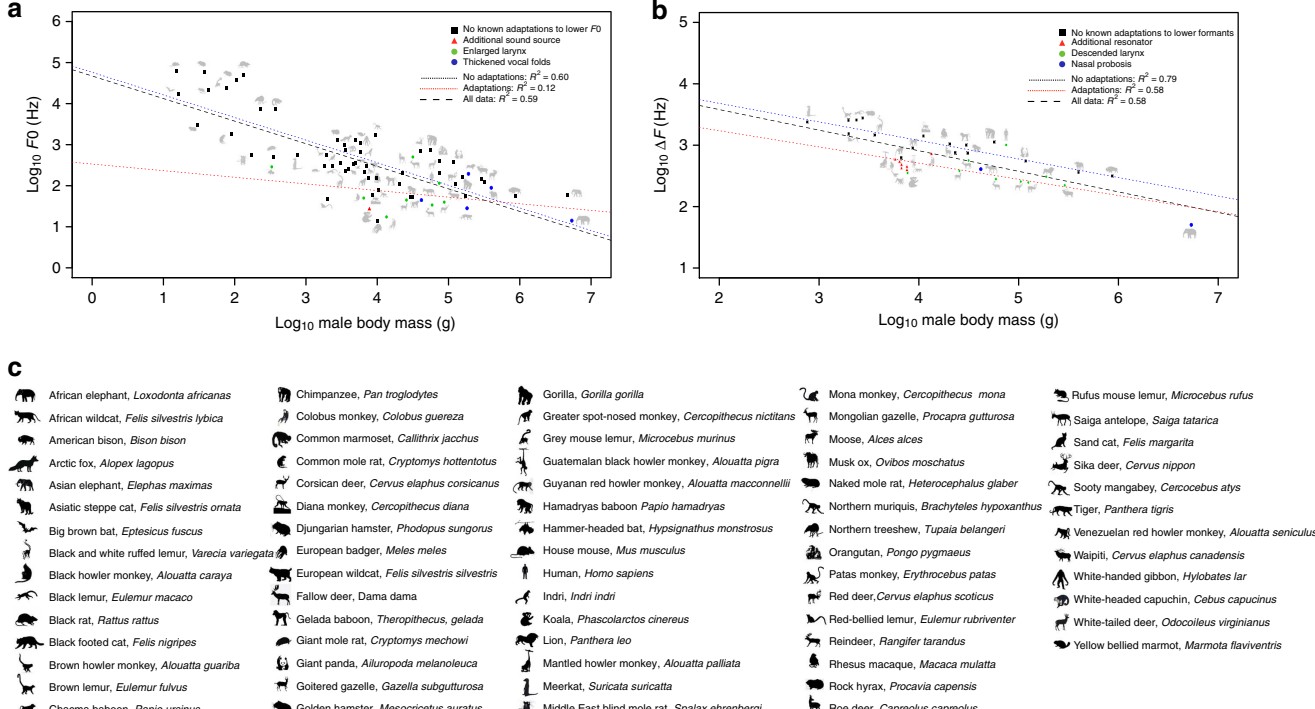

**Figure 1 | Relationship between male body mass and acoustic variables across terrestrial mammal species.** The scatterplots show the relationship between (**a**) $\log_{10}$ male body mass and $\log_{10} F0$ and (**b**) $\log_{10}$ male body mass and $\log_{10} \Delta F$. The dotted lines represent the slope and intercept of phylogenetic generalized least-squares regressions of $\log_{10}$ male body mass on $\log_{10} F0$ (**a**) and $\log_{10}$ male body mass and habitat on $\log_{10} \Delta F$. (**b**) In both plots, the blue dotted line shows the relationship between male body mass and acoustic features for species without known adaptations to lower frequency components of calls, the red dotted line shows the relationship between male body mass and acoustic features for species that are known to possess adaptations to lower frequency components of calls, and the black dotted line shows the relationship between male body mass and acoustic features for all the species in the data set (plot A: PGLS regression, $N = 67$, $P < 0.001$; plot B: PGLS regression, $N = 35$, $P < 0.001$). In both plots a black square indicates species with no known adaptations to lower frequency components. A red triangle denotes species with additional sound sources and resonators in plots A and B, respectively, a green circle signifies species with an enlarged larynx in plot A and species with a descended larynx in plot B, and a blue circle represents species with thickened vocal folds in plot A and a nasal proboscis in plot B. A key for the icons representing each of the mammal species is also provided in **c**.

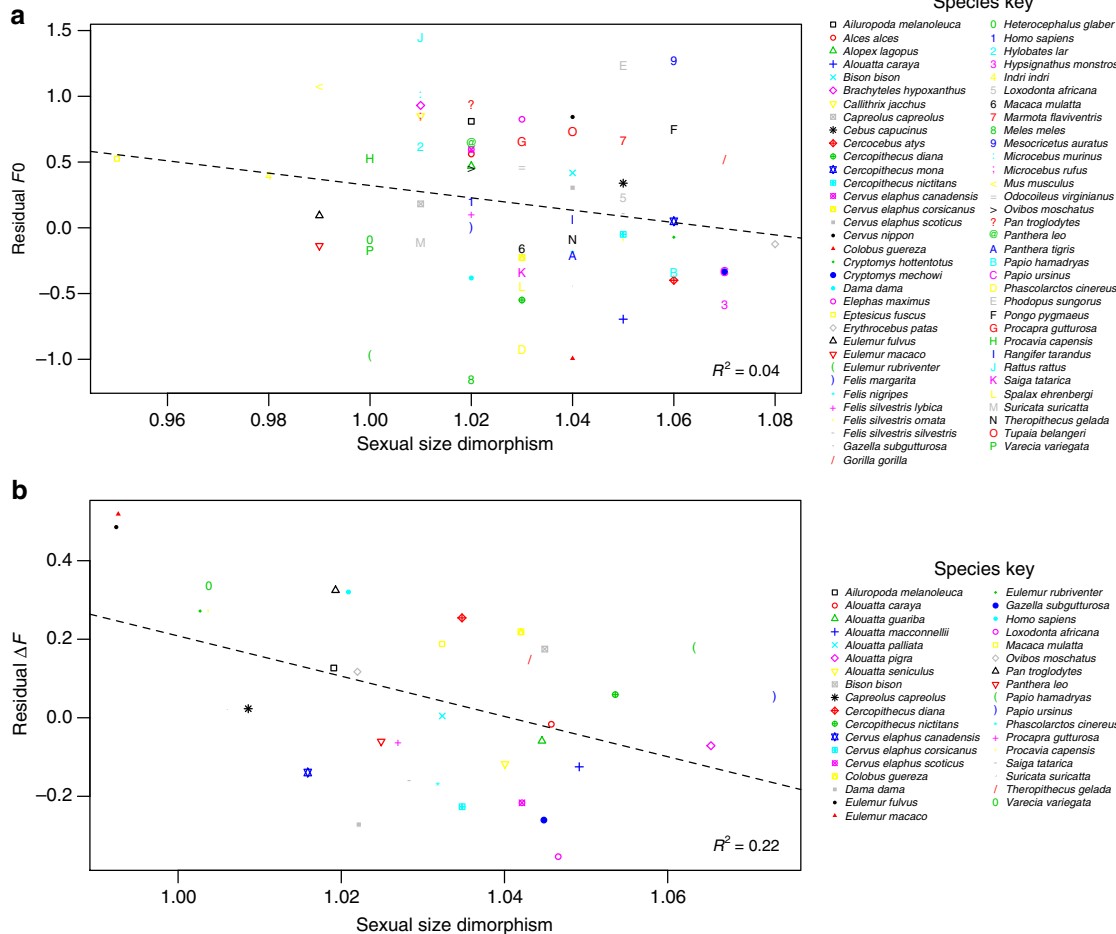

**Figure 2 | Relationship between sexual size dimorphism and acoustic variables across terrestrial mammal species.** The scatterplots show the relationship between (**a**) sexual size dimorphism and residual $F0$, and (**b**) sexual size dimorphism and residual $\Delta F$. Residual $F0$ and $\Delta F$ refer to the residuals obtained from PGLS regressions of $\log_{10}$ male body mass on $\log_{10}$ $F0$ and $\log_{10}$ male body mass and habitat on $\log_{10}$ $\Delta F$, respectively. For each plot, the dotted line represents the slope and intercept of the PGLS model regressions (plot A: $N = 67$, $P = 0.119$; plot B: $N = 35$, $P = 0.006$). $R^2$ values are given in the bottom right-hand corner.

Our phylogenetic analysis also reveals that male mammals with relatively large testes produced calls with higher $\Delta F$, suggesting that pre-copulatory sexual selection pressures to acoustically exaggerate body size are relaxed in species where sperm competition predominates. This result confirms the evolutionary trade-off between acoustic size exaggeration and testes size revealed by a recent study of sexual calls in howler monkeys[20]. The fact that $F0$ is also lower in species with relatively smaller testes is consistent with previous observations that, while lower $F0$ may not function as a reliable cue to body size within mammal species, it can indicate higher testosterone levels[42,43], threat potential[44] and dominance[45,46], and hence, remains an important, sexually selected component of pre-copulatory signalling in mammals. Indeed, recent findings in anthropoid primates show how sexual dimorphism in $F0$ increases during evolutionary transitions towards polygyny and decreases during transitions towards monogamy[33], further emphasizing that $F0$ is a sexually selected component of mammal vocalizations.

Finally, our comparative investigation provides a useful background for understanding the selection pressures contingent on our own species' vocal communication. Although male humans do not appear to possess an exclusively sexual call, it is now well established that $F0$ and formants are sexually selected components of the male voice that play a role in mate choice[47,48] and intra-sexual competition[44,46]. However, unlike other

primates, adult humans have a descended larynx that results in a disproportionately long pharyngeal cavity[1]. Moreover, a secondary descent of the larynx that only affects adult men at puberty, and enables them to produce even lower formant frequencies, has been attributed to sexual selection for size exaggeration[49]. Yet surprisingly, the observed $\Delta F$ of male humans is quite far above, rather than below the value predicted from the acoustic allometry (Fig. 1b), indicating that the human male vocal tract is in fact shorter than expected for a male terrestrial mammal that weighs around 75 kg (the average weight of a male human[50]). We suggest that selection pressures to decrease facial size may have counter-balanced sexual selection pressures to exaggerate apparent body size, and resulted in the relative overall shortening of the human vocal tract revealed by our comparative investigation. Indeed, selection pressures linked to speech production, thermoregulation or locomotion[51] and facilitated by tool use and meat eating[52], are generally assumed to have led to the 1:1 ratio of the oral cavity relative to the pharyngeal cavity that is considered to be a crucial prerequisite for the evolution of complex speech articulation[1,53]. Our phylogenetically controlled investigation, therefore, not only reveals how sexual selection for acoustic size exaggeration drives the anatomical and acoustical diversification of terrestrial mammal vocal communication systems, but also highlights the importance of the comparative approach for providing the background necessary to fully

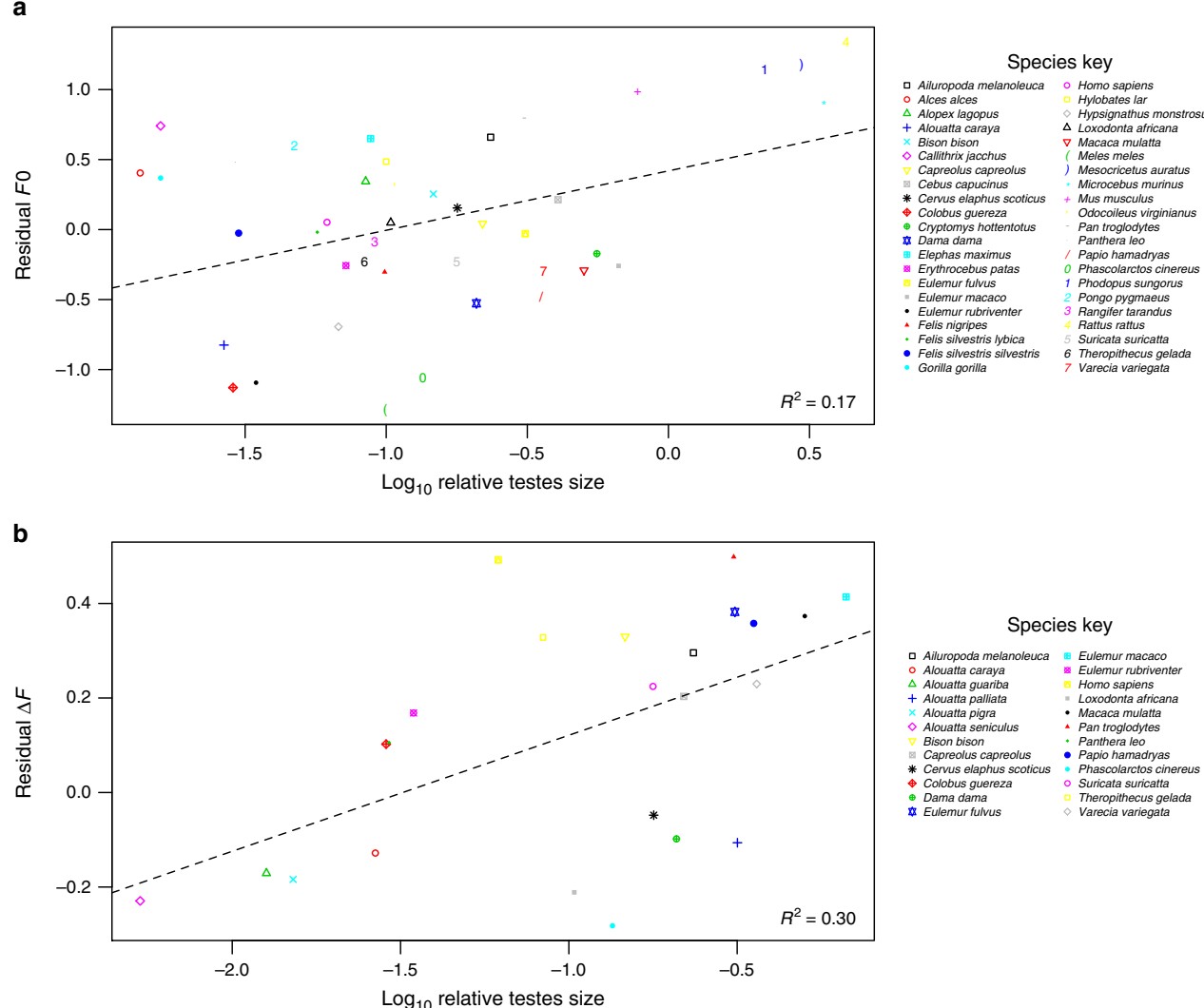

**Figure 3 | Relationship between relative testes size and acoustic variables across terrestrial mammal species.** The scatterplots show the relationship between (**a**) $log_{10}$ relative testes size and residual $F0$, and (**b**) $log_{10}$ relative testes size and residual $\Delta F$. Residual $F0$ and $\Delta F$ refer to the residuals obtained from PGLS regressions of $log_{10}$ body mass on $log_{10}$ $F0$ and $log_{10}$ $\Delta F$, respectively. The dotted line represents the slope and intercept of the PGLS model regressions (plot A: $N = 42$, $P = 0.017$; plot B: $N = 24$, $P < 0.001$). $R^2$ values are given in the bottom right-hand corner.

understand the origins and evolution of our own species' vocal apparatus[1].

## Methods

**Data sources.** To test our hypotheses, we collated acoustic data on mean $F0$ from 67 male species across 52 genera, and mean formant frequency values from 35 male species across 25 genera from the literature (Supplementary Table 7). We restricted the data set to adult terrestrial mammals and noted whether vocalizations function as sexual calls (that is, those that are purported to have functional relevance during intra-sexual or inter-sexual assessment). This allowed us to enter call-type (sexual or nonsexual) as a covariate in the analysis, and control for any differences in the acoustic structure generated by males using different modes of sound production and/or adopting different calling postures exclusively in sexual calls (such as the use of non-laryngeal sources and/or vocal tract elongation by laryngeal retraction or neck stretching). Humans were included on the basis that $F0$ and formants in the male voice have also been shaped by sexual selection[46–48].

In cases where mean $F0$ and formant frequency values were not directly reported in papers, the acoustic values were obtained by contacting the lead authors of the respective studies. For five species, mean $F0$ was estimated by taking the average of the minimum and maximum reported values[3,32]. To calculate formant frequency spacing ($\Delta F$), we used the first two to nine formant frequencies (mean = 5) and the regression method of Reby & McComb[54], in which the formant frequency values are plotted against those that would be expected if the vocal tract was a straight uniform tube closed at one end (the glottis) and open at the other

(the mouth). This regression method is an accurate way to estimate $\Delta F$ in species with unevenly spaced formants (as is commonly the case in mammals[17,22,54]).

Data on male acoustic features and body mass were obtained from the same published source for 39 out of 67 taxa for the $F0$ data and 15 out of 35 taxa for the formant data. In addition, because the physical environment also shapes the acoustic features of vocal signals[32,38,39], we collected data on the typical habitat for each of the species in our comparative analyses from the Encyclopaedia of Life website (http://eol.org/) to control for this factor in the analyses. We also collected data on the mating system of each species in the analysis from the Animal Diversity Website (http://animaldiversity.org/). If body weight data was not available from the acoustic studies we referred to the CRC handbook of mammalian body masses[55] and the PANTHERIA v.1 database[56]. We did not collect acoustic and body weight data for farmed or domestic animals (for example, cats, dogs, horses, sheep, goats) that are often intensely bred and therefore subject to strong artificial selection.

The degree of sexual size dimorphism was used as an indicator of the intensity of sexual selection pressures acting on male body size in a given species[57,58] with greater values indicative of selection pressures for larger male body size. Sexual size dimorphism was calculated for each species by dividing $log_{10}$ male body weight by $log_{10}$ female body weight (to convert a cubic measure to a linear measure of size[57,58]). Body mass data were taken from several sources (Supplementary Table 7); however, care was always taken to match male and female body mass data from the same population when calculating sexual size dimorphism. Relative testis size was used as an index of post-copulatory sexual selection pressures[20,59,60]. Post-copulatory sexual selection is prevalent in promiscuous species that live at high population densities and typically manifests itself as sperm competition,

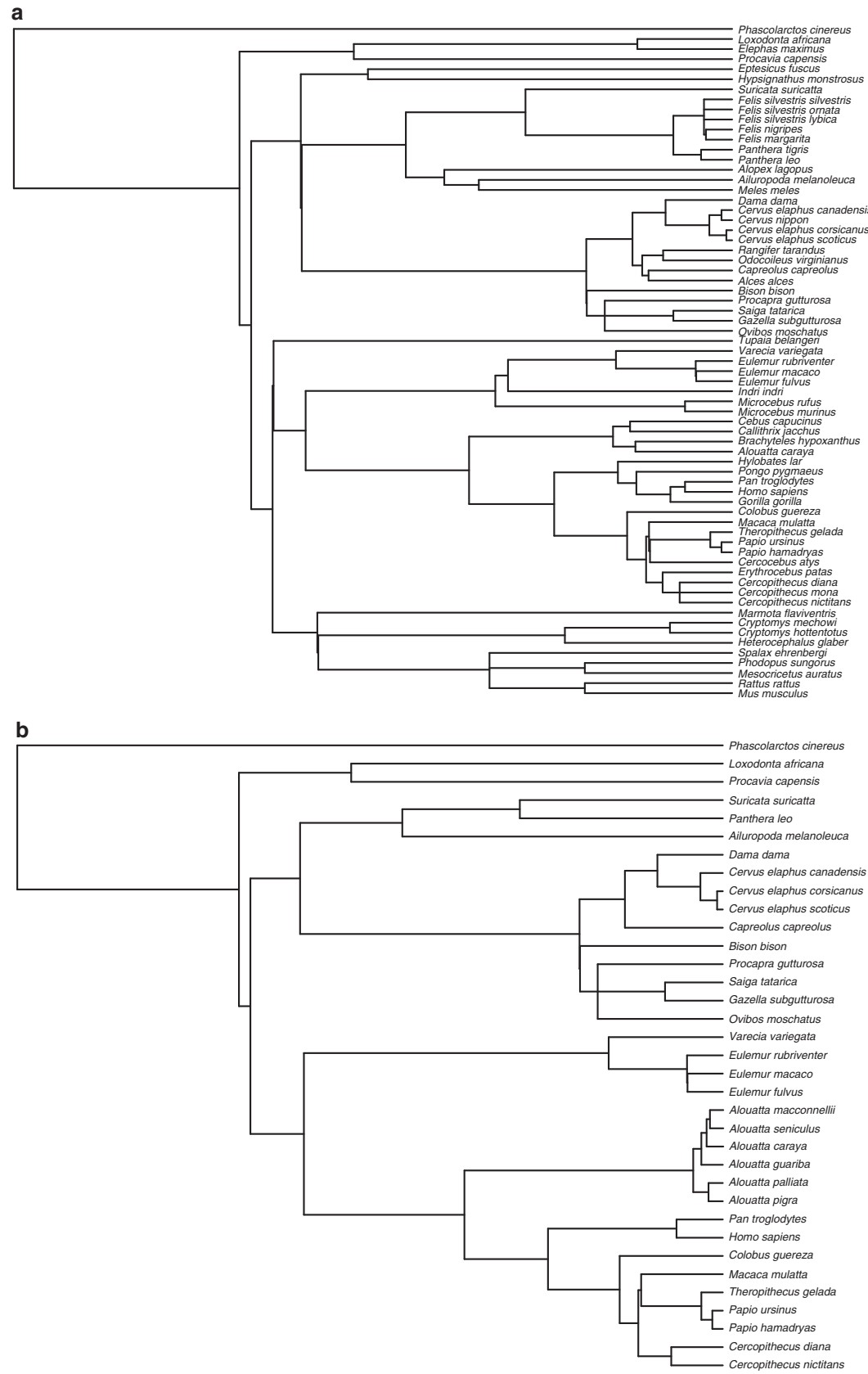

**Figure 4 | The phylogenies used to control for shared ancestry between different species.** PGLS regressions testing the effect of size dimorphism and relative testes size on *F*0 used the phylogeny in **a**; and those testing the effect of size dimorphism and relative testes size on Δ*F* used the phylogeny in **b**.

which in turn leads to larger male testes relative to overall body size[61]. Relative testes size is thus assumed to be a reliable index of the degree of sperm competition experienced by males within a species[20,59,60]. We calculated relative testes size for each species as the total mass of both testis in grams divided by the overall body mass in grams, rather than generating residuals of male testes mass regressed on body mass across taxa[59,62]. Male body and testes mass data from the same population were collected for 42 species for the $F0$ analyses and 24 species for the $\Delta F$ analyses (Supplementary Table 7). In five cases where data on testes mass were not directly available, the mass in grams was calculated by multiplying the volume in mm$^3$ by 1.02 (ref. 60).

**Statistical analyses.** Due to shared phylogenetic history, data from different species cannot be treated as statistically independent[20,57,58,63]. Accordingly, we conducted PGLS regressions using the gls function (nlme package) in $R$[64] to test our hypotheses. To control for the confounding effects of shared phylogenetic ancestry we used untransformed branch lengths and splitting dates from a recent molecular phylogeny of mammals[65]. Additional molecular phylogenies were used to improve resolution within the *Cervinae*[66], *Alouatta*[67] and *Cercopithecinae*[68] species. Figure 4 depicts the phylogenies used to control for shared ancestry among species in each of the separate analyses.

In addition, for each formal hypothesis we computed five PGLS regression models that were designed to test a different evolutionary scenario, and chose the most parsimonious model with the lowest Akaike Information Criterion statistic corrected for sample size (AICc)[37,69]. The different models were an Ornstein–Uhlenbeck (OU) model of evolution, a non-phylogenetic ordinary least-squares (OLS) model, a pure Brownian motion (BM) model, and two restricted maximum-likelihood (REML) Brownian motion models that allow parameters to vary with the strength of the phylogenetic signal, a Brownian motion + Pagel's lambda (BM + λ) and a Brownian motion + Grafen's rho (BM + ρ) model. The OLS model assumes phylogenetic independence, the BM model assumes a Brownian motion model of trait evolution (or pure Random Walk), the OU model uses alpha (α) to test the strength of stabilizing selection: α = 0 is equivalent to pure Brownian motion and larger values of α indicate stronger stabilizing selection, the BM + λ model allows us to test if the best model falls between pure Brownian motion (λ = 1) and phylogenetic independence (λ = 0), and the BM + ρ model tests the rate of evolutionary change, with ρ < 1 indicating relatively more gradual recent evolution, ρ > 1 relatively faster recent evolution, and ρ = 0 indicates a star phylogeny, generated by a recent population expansion event from a common ancestor.

For each PGLS regression, the dependent variable was the acoustic measure ($\log_{10} F0$ or $\log_{10} \Delta F$). $\log_{10}$ transformed male body mass (in grams) was entered as a covariate to control for body size differences across taxa, and the species-typical habitat (arboreal versus terrestrial), the call-type from which the acoustic data was derived (sexual or nonsexual), and the mating system for each species (monogamous, polygynous, polyandrous, promiscuous or variable) were also entered into a global PGLS model to control for these factors. For each formal hypothesis, we then used the 'dredge' function in $R$ (MuMIn' package) to iterate through all variable combinations in the global model to explain variation in $\log_{10} F0$ and $\log_{10} \Delta F$, and chose the model with the lowest AICc value[69]. The acoustic variables $F0$ and $\Delta F$, and relative testes size were $\log_{10}$ transformed to achieve a normal data distribution. All other variables were normally distributed.

**Data availability.** The data that support the findings of this study are available in Supplementary Table 7 and also from the corresponding author upon request.

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

## Acknowledgements

We thank Claudia Barelli, Jack Bradbury, Tecumseh Fitch, Roland Frey, Marco Gamba, Tara Harris, Lee Koren, Simon Townsend, Ilya Volodin, lena Volodin and Megan Wyman for providing unpublished data and/or clarification on some of the acoustic data.

## Author contributions

B.D.C. and D.R. conceived and designed the study; B.D.C. and D.R. wrote the manuscript; B.D.C. collated the data and conducted the analyses.

## Additional information

**Competing financial interests:** The authors declare no competing financial interests.

