## [Peer Review File · Nature Communications]

Reviewers' comments:

Reviewer #1 (Remarks to the Author):

Review of: Charlton & Reby

Review by: Dan Blumstein

The authors conduct a set of comparative analyses to understand what acoustic attributes, precisely, covary with body size in mammalian sexual vocalizations. Specifically, they test the acoustic allometry hypothesis which becomes interesting because some species have evolved ways to produce lower-frequency vocalizations than would be expected based on body size alone. The authors correctly assert that this is a phylogenetically broad analysis that includes more genera of mammals than previously studied and is especially novel for this reason. They compare two specific acoustic features (F0 and ΔF) and discuss their results from different, complementary, perspectives (evolutionary, functional, and proximate). Their understanding of mechanisms underlying sound production is deep and I learned a lot reading this discussion. Their conclusion, while not surprising (sexual selection for body size is a key driver of acoustic diversity in mammals) was well-justified given the taxonomic breadth of the analysis. Overall, I love the idea, the justification of the analysis, and the results, but I have some lingering concerns about the details of the analyses that should be cleared up before I feel comfortable recommending this paper for publication.

Fundamentally, I'm puzzled about why the authors elected to use residuals rather than fitting one PGLS model for each formal hypothesis with all variables (e.g., the go-to Garamszegi 2014 book). I believe that fitting a single model is generally preferred practice so it's somewhat surprising that they didn't do this. I don't have a problem plotting results as they did (figures are great!) but I'd like the statistical analyses to be a multivariate one that controls for these variables in one model (see the paper on the use and misuse of residuals...).

I'm also puzzled as to why the authors elected to not compare different models (with and without phylogenetic hypotheses and use model comparison to identify the best of those models). This is becoming preferred practice and regardless of whether there is a significant Pagel's lambda value, a non-phylogenetic model may still be the most parsimonious model. I encourage the authors to formally compare non-phylogenetic, with Brownian motion, punctuated, and perhaps OU models of evolution and report the best (I've done this in some previous work-e.g., Hensley et al. 2015 *Current Zoology* 61:773-780 but Ted Garland has done this in many of his recent papers and it's discussed in various places in the go-to Garamszegi 2014 book).

Thus, while I think this is a very well written paper with a potentially a comprehensive data set and an important conclusion, I'd like to see some new analyses here or a really good justification of why they made the analysis decisions that they have.

Minor stuff

Bibliography needs proofing.

Reviewer #2 (Remarks to the Author):

The evolution of acoustic size exaggeration in terrestrial mammals

Charlton and Reby

This is a very interesting and well-written manuscript, and should make an important contribution to the literature. It is the first study to examine F0 and ΔF across a diverse array of mammals, and has produced novel/important results. Although, the overall results for F0 and ΔF are not particularly surprising, nevertheless, it is important to show that these relationships are evident across mammals. I find the relationship between expected F0, ΔF , and reduced sperm competition intriguing, if it is robust. The difference between arboreal versus terrestrial species is also noteworthy and should be in the Abstract if the word limit allows.

The authors should consider the findings of a very recent related study in Proc B, by Puts et al. "Sexual selection on male vocal fundamental frequency in humans and other anthropoids". It would be good to try to integrate the F0 data from other anthropoids into the current study. Particularly because the results are somewhat contrary. "Here we show across anthropoids that sexual dimorphism in fundamental frequency (F0) increased during evolutionary transitions towards polygyny, and decreased during transitions towards monogamy."

Some of the overall message in the Abstract is unclear: "sexual selection does not favour the use of F0 as an acoustic size exaggerator" is followed by "males produced sexual calls with lower than expected F0 and FFS in mating systems characterised by reduced sperm competition". Either F0 is, or is not, linked to sexual selection. Which is it?

The authors should have included line numbers so that commenting on the ms is made easier.

Abstract. Line 1, remove "will". Line 3, remove "should".

Introduction, paragraph 3. State clearly the number of species, not simply 8 orders.

Results, Line 1. Repeat species sample size after "across species".

Results, paragraph 2, line 5. Revise "male species".

Discussion, line 1. The word "trend" is often used to refer to statistically non-significant results that approach significance. Revise.

Discussion, paragraph 5. This paragraph needs to be revised by taking into consideration the study by Puts et al. that has just been published.

"Yet surprisingly, the observed ΔF of male humans is quite far above, rather than below the value predicted from the acoustic allometry, indicating that the human male vocal tract is in fact shorter than expected for a male terrestrial mammal that weighs around 75 kg (the average weight of a male human 56)." Give the diversity of sizes across human ethnicities, is this relationship robust?

Reviewer #3 (Remarks to the Author):

This is a theoretically interesting paper that is clearly written, and which explores a valuable data set. The authors found that male terrestrial mammals evolve lower "sexual" vocalisations than predicted from body size in terrestrial vs. arboreal species and in species with greater levels of body size dimorphism. The latter relationship applied to formant frequencies but not fundamental frequency. Both acoustic measures were negatively related to relative testes size, a measure of the level of sperm competition. The topic will be of broad interest. However, the present data and analyses do not support the authors' conclusions that "male terrestrial mammals produce mating calls with lower ΔF than expected for their size in mating systems with sexual selection pressures for large male body size.... [and] that sexual selection does not favour the use of F0 as an acoustic size exaggerator." The reasons are as follows:

1. One obtains an imprecise reflection of the influence of sexual selection by looking at fundamental and formant frequencies of vocalisations relative to body size in males alone. This is because many selection pressures influence acoustic frequencies across species. Some of these are sexual selection pressures, while others are ecological (e.g., arboreal vs. terrestrial habitat, as the authors found). One can better estimate the contributions of sexual selection by comparing the sexes, as females are likely to have experienced relatively more ecological than sexual selection and can provide a within-species "control" for ecological (or non-sexual social) selection pressures. Thus, when examining the evolution of male frequencies as a function of sexual selection, it is important to consider how male frequencies contrast with those of females—that is, to consider sexual dimorphism in vocal frequencies.

The parallel here is male body mass. One does not merely assume that male mass increases with mating competition, because many other factors influence body mass (predation, diet, climate, and so forth). Rather, one compares body mass *dimorphism* to the intensity of mating competition. For example, male green sea turtles are huge not because of intense sexual selection for size in males, but rather because of ecological selection pressures that produce large bodies in both sexes (and indeed greater size in females). Similarly, acoustic dimorphisms (not acoustic frequencies in one sex alone) should be related to the intensity of mating competition. This conclusion is clear in the present results. For example, human male vocal frequencies were found to be either at expected levels for body mass (F0, all data) or considerably higher than expected for a species with male vocal adaptations, suggesting very weak sexual selection on male vocal frequencies. Yet this conclusion is probably incorrect, as the authors point out: copious evidence points to a strong role of sexual selection on human male voices. When one instead considers F0 dimorphism, this becomes clear; a recent study by Puts et al. (2016, see below) found that humans exhibit the greatest F0 dimorphism of any ape. Thus, although men may exhibit average-to-high vocal frequencies for a primate of their size, this may reflect non-sexual selection pressures (e.g., articulatory clarity in speech) rather than low sexual selection. When one examines acoustic sexual dimorphisms, the influence of sexual selection is apparent.

2. The authors used sexual size dimorphism "as an indicator of the intensity of sexual selection pressures acting on male body size in a given species". This a reasonable decision for measuring sexual selection pressures acting specifically *on male body size*, but one cannot extrapolate from sexual selection on body size to the intensity of sexual selection more broadly (Plavcan, 2004; 2011; 2012), as the authors do, e.g., "When investigating the effect of sexual selection, we found that sexual size dimorphism did not predict F0 across taxa..."

There are two main reasons for this. First, size dimorphism is only a modest indicator of one form mating competition: overt aggression. The frequency and intensity of male-male agonism explained only 48% of the variation in sexual size dimorphism in a sample of 128 anthropoid primate species (Plavcan, 2012). Second, mating competition takes many forms other than fighting. Because there are numerous

forms of mating competition (e.g., sperm competition, scramble competition) that do not produce large size dimorphism, sexual size dimorphism is a poor proxy for the intensity of mating competition overall (e.g., Plavcan, 2012). Thus, another measure of the intensity of sexual selection should be used, such as breeding system.

3. A recent paper by Puts and colleagues (2016) is highly relevant to the present work and should be discussed in the context of the present study. These authors explored the evolution of F0 sexual dimorphism in relation to breeding system, habitat, and body size dimorphism in anthropoid primates. Of particular relevance is their finding relating sexual F0 dimorphism to breeding system and body size dimorphism: Low male (relative to female) vocalisations related negatively to size dimorphism, once breeding system was controlled. They hypothesized that this is because the function of low-frequency male vocalizations is precisely for **avoiding** fights (via size exaggeration). Thus, on the one hand, inter- or intrasexual mating competition may produce both acoustic and body size dimorphisms. Meanwhile, on the other hand, because large body size helps males win fights, and low vocalisations may help males avoid fights, the two may be inversely related when the overall intensity of male mating competition is controlled. These results are also similar in principle to those of Dunn et al. (2015; cited in the present work), who found that male howler monkeys do not evolve acoustic anatomical dimorphisms when their mating competition predominantly takes a form other than direct fighting (sperm competition, in the case of howlers). Indeed, in the present work, relatively low frequency (in F0 and ΔF) male calls occurred in species with relatively small testes, suggesting that the form of mating competition rather than its overall intensity influences the evolution of male calls, and cautioning against using a (somewhat weak) measure of male agonism as a proxy for overall mating competition.

4. The authors collected acoustic data only from vocalisations utilized in presumptive "sexual" calls. However, it should be noted that the functions of calls are not known with certainty, and other "non-sexual" calls may in fact be relevant to mating competition. Moreover, given that sexual selection pressures produce low frequency vocalisations by shaping the underlying anatomy (larynges, vocal sacs, etc.), the effects of sexual selection should be observable across the range of vocalisation types produced in a species. Finally, in order to measure sexual dimorphism in vocal frequencies (see above), it will be necessary to measure "non-sexual" calls, as females do not produce "sexual" calls in some species.

References

Plavcan, J. M. (2004). Sexual selection, measures of sexual selection, and sexual dimorphism in primates. In P. M. Kappeler & C. P. van Schaik (Eds.), *Sexual selection in primates: New and comparative perspectives* (pp. 230-252). Cambridge: Cambridge University Press.

Plavcan, J. M. (2011). Understanding dimorphism as a function of changes in male and female traits. *Evolutionary Anthropology*, 20, 143-155.

Plavcan, J. M. (2012). Sexual size dimorphism, canine dimorphism, and male-male competition in primates: where do humans fit in? *Hum Nat*, 23(1), 45-67. doi: 10.1007/s12110-012-9130-3

Puts, D. A., Hill, A. K., Bailey, D. H., Walker, R. S., Rendall, D., Wheatley, J. R., . . . Ramos-Fernandez, G. (2016). Sexual selection on male vocal fundamental frequency in humans and other anthropoids. *Proc Biol Sci*, 283(1829). doi: 10.1098/rspb.2015.2830

Reviewer #1 (Dan Blumstein)

The authors conduct a set of comparative analyses to understand what acoustic attributes, precisely, covary with body size in mammalian sexual vocalizations. Specifically, they test the acoustic allometry hypothesis which becomes interesting because some species have evolved ways to produce lower-frequency vocalizations than would be expected based on body size alone. The authors correctly assert that this is a phylogenetically broad analysis that includes more genera of mammals than previously studied and is especially novel for this reason. They compare two specific acoustic features (F_0 and ΔF) and discuss their results from different, complementary, perspectives (evolutionary, functional, and proximate). Their understanding of mechanisms underlying sound production is deep and I learned a lot reading this discussion. Their conclusion, while not surprising (sexual selection for body size is a key driver of acoustic diversity in mammals) was well-justified given the taxonomic breadth of the analysis. Overall, I love the idea, the justification of the analysis, and the results, but I have some lingering concerns about the details of the analyses that should be cleared up before I feel comfortable recommending this paper for publication.

We are grateful to Dan Blumstein for his encouraging and constructive comments.

Fundamentally, I'm puzzled about why the authors elected to use residuals rather than fitting one PGLS model for each formal hypothesis with all variables (e.g., the go-to Garamszegi 2014 book). I believe that fitting a single model is generally preferred practice so it's somewhat surprising that they didn't do this. I don't have a problem plotting results

as they did (figures are great!) but I'd like the statistical analyses to be a multivariate one that controls for these variables in one model (see the paper on the use and misuse of residuals...).

We now run multivariate PGLS models for each of our formal hypotheses and no longer use residuals. In each PGLS the dependent variable is the acoustic measure (\log_{10} F0 or \log_{10} formant spacing) and \log_{10} male body weight is entered as a covariate in all models to control for body size differences across taxa. As before, the independent variable was either male size dimorphism or relative testes size depending on whether we were examining the effect of pre-copulatory or postcopulatory sexual selection pressures on male acoustics. In addition, we now use a model selection procedure based on the AICc for all the PGLS regressions. In each case the global model includes male body mass, habitat type (terrestrial versus arboreal), call-type (sexual or nonsexual), and mating system (monogamous, polygynous, polyandrous, promiscuous and variable) as covariates, and the model with the lowest AICc value is chosen as the best supported model (Burnham, K. D. & Anderson, D. R. *Model selection and multimodal inference: a practical information-theoretic approach*. 2nd edn, (Springer-Verlag, 1998)). Please note that we have now considerably increased our dataset by including nonsexual calls (in line with Reviewer 3's suggestion).

All of the above information is now included in the manuscript and the model selection is detailed in the supplementary tables S2-S7.

I'm also puzzled as to why the authors elected to not compare different models (with and without phylogenetic hypotheses and use model comparison to identify the best of those models). This is becoming preferred practice and regardless of whether there is a significant Pagel's lambda value, a non-phylogenetic model may still be the most parsimonious model. I encourage the authors to formally compare non-phylogenetic, with Brownian motion, punctuated, and perhaps OU models of evolution and report the best (I've done this in some previous work-e.g., Hensley et al. 2015 *Current Zoology* 61:773-780 but Ted Garland has done this in many of his recent papers and it's discussed in various places in the go-to Garamszegi 2014 book).

We now formally compare five different evolutionary scenarios and choose the most parsimonious PGLS regression model with the lowest Akaike Information Criterion statistic corrected for sample size (AICc) *sensu* Hensley et al. 2015 (model selection is reported in supplementary Tables S2-S7).

The models compared are:

1. OLS (non-phylogenetic model)
2. Brownian motion ("corBrownian" correlation structure in R)
3. Brownian motion + Pagel's lambda ("corPagel" correlation structure in R)
4. Brownian motion + Grafen's rho ("corGrafen" correlation structure in R)
5. Ornstein-Uhlenbeck ("corMartins" correlation structure in R)

The PGLS regressions are now computed using the `gls` function (nlme package) in R, which allows us to include an Ornstein-Uhlenbeck (OU) model of evolution, along with a non-phylogenetic model, a pure Brownian motion model, and two maximum-likelihood Brownian motion models that allow parameters to vary with the strength of the phylogenetic signal. It is important to appreciate that maximum likelihood estimates can differ significantly from 0. The maximum-likelihood Brownian motion models are free to compute the parameter value that best fits the data, rather than forcing the parameters to test for the extreme scenarios (e.g. species specialization versus adaptive radiation, gradual versus punctuated evolution).

The Brownian motion + Pagel's lambda (λ) model allows us to test if the best model falls between pure Brownian motion ($\lambda = 1$) and phylogenetic independence ($\lambda = 0$). Grafen's rho (ρ) tests the rate of evolutionary change, with $\rho < 1$ indicating relatively more gradual recent evolution (branches leading to the tips of the phylogeny are stretched), $\rho > 1$ relatively faster recent evolution (branches leading to the root are stretched), and $\rho = 0$ indicates a star phylogeny in which many short branches are connected at the internal node, representing a recent population expansion event from a common ancestor. Finally, the OU model uses alpha (α) to test the strength of stabilizing selection: $\alpha = 0$ is equivalent to pure Brownian motion and larger values of α indicate stronger selection.

Thus, while I think this is a very well written paper with a potentially a comprehensive data set and an important conclusion, I'd like to see some new analyses here or a really good justification of why they made the analysis decisions that they have.

Minor stuff

Bibliography needs proofing.

This has now been done.

Reviewer #2 (Remarks to the Author):

#NCOMMS-16-08040 - The evolution of acoustic size exaggeration in terrestrial mammals

Charlton and Reby

This is a very interesting and well-written manuscript, and should make an important contribution to the literature. It is the first study to examine F_0 and ΔF across a diverse array of mammals, and has produced novel/important results. Although, the overall results for F_0 and ΔF are not particularly surprising, nevertheless, it is important to show that these relationships are evident across mammals. I find the relationship between expected F_0 , ΔF , and reduced sperm competition intriguing, if it is robust. The difference between arboreal versus terrestrial species is also noteworthy and should be in the Abstract if the word limit allows.

We would rather not report covariate results in the abstract as the word limit is

only 150 words.

The authors should consider the findings of a very recent related study in Proc B, by Puts et al. "Sexual selection on male vocal fundamental frequency in humans and other anthropoids". It would be good to try to integrate the F0 data from other anthropoids into the current study.

This excellent paper, and its data, only became available after we submitted our manuscript. We have now integrated data on F0 in seven additional species of primates (*Brachyteles hypoxanthus*, *Callithrix jacchus*, *Cebus capucinus*, *Cercocebus atys*, *Cercopithecus mona*, *Erythrocebus patas* *Macaca mulatta*) reported in Puts et al into the current study (they did not collect data on formants). To remain consistent with the rest of our dataset we only include F0 data derived from a single call type, either a sexual call or a nonsexual call.

Particularly because the results are somewhat contrary. "Here we show across anthropoids that sexual dimorphism in fundamental frequency (F0) increased during evolutionary transitions towards polygyny, and decreased during transitions towards monogamy."

It is difficult to directly compare our results with those of Puts *et al* because we do not look at how vocal dimorphism changes across different mating systems. Although our results do not support the hypothesis that F0 is used as an acoustic size exaggerator, male size dimorphism and F0 are negatively correlated (though not statistically significant: see Fig. 2A). In that respect, because we find lower F0 in systems with stronger sexual selection (in this case for larger male body size) our results are not contrary to Puts *et al*. If we had used a more general index of sexual selection pressures, such as mating system, and only compared monogamous versus polygynous species (*sensu* Puts *et al*), we might have revealed a statistically significant difference between these two classes of mating system. This was not, however, the aim of the current study, nor do we consider our sample of monogamous species large enough to merit such an analysis.

Some of the overall message in the Abstract is unclear: "sexual selection does not favour the use of F0 as an acoustic size exaggerator" is followed by "males produced sexual calls with lower than expected F0 and FFS in mating systems characterised by reduced sperm competition". Either F0 is, or is not, linked to sexual selection. Which is it?

We do not feel that these statements are contradictory, but appreciate that they could be clearer.

In response to this comment we have now removed the statement "sexual selection does not favour the use of F0 as an acoustic size exaggerator" from the abstract.

We do not discount the fact that F0 is under sexual selection. As extensively

indicated by the research of Puts and colleagues, including the recent aforementioned publication, we believe that F0 is likely to be sexually selected as a cue to maleness/androgens/dominance, hence the relaxation observed in species with mating systems characterised by increased sperm competition with less pre-copulatory competition.

We now make this clear on lines 180-187:

“The fact that F0 was also lower in species with relatively smaller testes is consistent with previous observations that, while lower F0 may not function as a reliable cue to body size within mammal species, it can indicate higher testosterone levels⁴⁹⁻⁵¹, threat potential and dominance^{29,52}, and hence, remains an important, sexually-selected component of pre-copulatory signalling in mammals. Indeed, recent findings in anthropoid primates show how sexual dimorphism in F0 increases during evolutionary transitions towards polygyny, and decreases during transitions towards monogamy³³, further emphasizing that F0 is a sexually selected component of mammal vocalisations.

The authors should have included line numbers so that commenting on the ms is made easier.

We apologize for this oversight and the inconvenience it has caused. The revised version now contains line numbers.

Abstract. Line 1, remove "will". Line 3, remove "should".

Done.

Introduction, paragraph 3. State clearly the number of species, not simply 8 orders.

We now state: “In this paper we provide the first phylogenetically controlled comparative examination of the selection pressures that lead to acoustic size exaggeration across nine orders and 72 species of terrestrial mammals.”

Results, Line 1. Repeat species sample size after "across species".

The sample sizes are given on the figures, at the start of the methods section, and are implicit in the degrees of freedom. Therefore we do not see the need to repeat them consistently throughout the results section. We do now state the sample sizes for the analysis of relative testes size versus acoustics in the results section though, to illustrate that this was a reduced dataset of species for which acoustic and testes size data were available.

Results, paragraph 2, line 5. Revise "male species".

We have changed “male species” to “males”.

Discussion, line 1. The word "trend" is often used to refer to statistically non-significant results that approach significance. Revise.

We have now changed “trends” to “results”.

Discussion, paragraph 5. This paragraph needs to be revised by taking into consideration the study by Puts et al. that has just been published.

We now state the following here:

“Indeed, recent findings in anthropoid primates show how sexual dimorphism in F0 increases during evolutionary transitions towards polygyny, and decreases during transitions towards monogamy³³, further emphasizing that F0 is a sexually selected component of mammal vocalisations.”

"Yet surprisingly, the observed ΔF of male humans is quite far above, rather than below the value predicted from the acoustic allometry, indicating that the human male vocal tract is in fact shorter than expected for a male terrestrial mammal that weighs around 75 kg (the average weight of a male human 56)." Given the diversity of sizes across human ethnicities, is this relationship robust?

The human data is taken from Pisanski et al 2016 (Animal Behaviour, 112, 13-22.). In this study, the authors obtained acoustic and body weight data from a very large sample of 296 Scottish and Canadian men. While data on more ethnicities, including relatively small or relatively large bodied populations would be interesting to consider, we believe that these data issued from a large sample of medium sized populations provide a representative index for humans. The data are also taken from the very populations in which the functions of nonverbal vocal components in sexual selection have been most extensively studied.

Reviewer #3 (Remarks to the Author):

This is a theoretically interesting paper that is clearly written, and which explores a valuable data set. The authors found that male terrestrial mammals evolve lower "sexual" vocalisations than predicted from body size in terrestrial vs. arboreal species and in species with greater levels of body size dimorphism. The latter relationship applied to formant frequencies but not fundamental frequency. Both acoustic measures were negatively related to relative testes size, a measure of the level of sperm competition. The topic will be of broad interest.

We thank the reviewer for their encouraging comments and helpful suggestions. We feel that many of the comments arise from a misunderstanding due to a lack of clarity in our original manuscript. Indeed, we did not intend to model the effect of sexual selection on vocal communication in general, but more specifically the effect of sexual selection for large male body size on acoustic size exaggeration. We apologise for this confusion, and trust that our more specific wording largely addresses the issues raised by the reviewer. Below we respond to specific comments, and propose additional analyses where suitable. However, we are concerned not

to broaden the scope of our investigation too far from its intended objectives, or to present results supported by insufficient data.

However, the present data and analyses do not support the authors' conclusions that "male terrestrial mammals produce mating calls with lower ΔF than expected for their size in mating systems with sexual selection pressures for large male body size.... [and] that sexual selection does not favour the use of F0 as an acoustic size exaggerator." The reasons are as follows:

One obtains an imprecise reflection of the influence of sexual selection by looking at fundamental and formant frequencies of vocalisations relative to body size in males alone. This is because many selection pressures influence acoustic frequencies across species. Some of these are sexual selection pressures, while others are ecological (e.g., arboreal vs. terrestrial habitat, as the authors found).

We entirely agree with this statement. Ecological factors will certainly have an affect on the acoustic structure of mammal vocal signals. We found that arboreal species have lower formant frequency spacing than other terrestrial species, and include habitat type as a covariate in our global PGLS regression models to control for this factor.

One can better estimate the contributions of sexual selection by comparing the sexes, as females are likely to have experienced relatively more ecological than sexual selection and can provide a within-species "control" for ecological (or non-sexual social) selection pressures.

If female acoustics are likely to be subject to greater ecological/nonsexual selection pressures than males (as stated above), then comparing the acoustic structure of male and female vocalisations will not tightly control for any effect of sexual selection on male acoustic structure. Female calls would only constitute a useful control if the vocal signals of both sexes were subject to the same ecological selection pressures, which cannot be assumed.

Thus, when examining the evolution of male frequencies as a function of sexual selection, it is important to consider how male frequencies contrast with those of females-that is, to consider sexual dimorphism in vocal frequencies.

We respectfully disagree with this statement. Our investigation has a very specific primary objective: to see whether species with relatively larger males than females are also species where males acoustically exaggerate size by lowering call frequencies (F0 and ΔF) beyond that expected for their body size. Our findings show that males from more size dimorphic species also produce calls with disproportionately lower ΔF , and this can only be explained by selection pressures for relatively larger body size in males that have driven the evolution of calls with exaggerated cues to size supported by anatomical and/or behavioural adaptations (e.g. a lowered and/or mobile larynx). We

believe that this is the most parsimonious explanation for such a relationship, and cannot think of a plausible alternative scenario.

Furthermore, published data on male and female formant frequencies from the same species is only available for seven terrestrial mammals. This relatively low sample size (and lack of statistical power) is clearly not sufficient to adequately test a hypothesis. However, we were able to collect data on male and female F0 values for 36 species that could be used to create a figure in the supplementary material if it is deemed necessary by the Associate Editor.

The parallel here is male body mass. One does not merely assume that male mass increases with mating competition, because many other factors influence body mass (predation, diet, climate, and so forth). Rather, one compares body mass *dimorphism* to the intensity of mating competition. For example, male green sea turtles are huge not because of intense sexual selection for size in males, but rather because of ecological selection pressures that produce large bodies in both sexes (and indeed greater size in females). Similarly, acoustic dimorphisms (not acoustic frequencies in one sex alone) should be related to the intensity of mating competition.

We entirely agree with this statement, and this is precisely why we use size dimorphism (rather than male size) as our index of sexual selection for body size. However, it is unnecessary to extend this approach to our acoustic index of size exaggeration. Indeed, if we take the example of the green sea turtle, the departure of acoustic components from values that are expected from males of this (very large) size would not be affected by the absolute size of males (and therefore ecological pressures), but indeed reflect pressure to exaggerate acoustic size.

This conclusion is clear in the present results. For example, human male vocal frequencies were found to be either at expected levels for body mass (F0, all data) or considerably higher than expected for a species with male vocal adaptations, suggesting very weak sexual selection on male vocal frequencies. Yet this conclusion is probably incorrect, as the authors point out: copious evidence points to a strong role of sexual selection on human male voices. When one instead considers F0 dimorphism, this becomes clear; a recent study by Puts et al. (2016, see below) found that humans exhibit the greatest F0 dimorphism of any ape. Thus, although men may exhibit average-to-high vocal frequencies for a primate of their size, this may reflect non-sexual selection pressures (e.g., articulatory clarity in speech) rather than low sexual selection. When one examines acoustic sexual dimorphisms, the influence of sexual selection is apparent.

We agree with the referee and already discuss how natural selection pressures on speech production are likely to have countered sexual selection pressures in humans on lines 200-207 of the manuscript. Here we explain how the relatively high frequency values observed in humans, despite

evidence of vocal dimorphism and sexual selection for low frequencies in males, may reflect natural selection pressures linked to speech production:

“We suggest that selection pressures to decrease facial size may have counter-balanced sexual selection pressures to exaggerate apparent body size, and resulted in the relative overall shortening of the human vocal tract revealed by our comparative investigation. Indeed, selection pressures linked to speech production, thermoregulation or locomotion⁵⁷ and facilitated by tool use and meat eating⁵⁸, are generally assumed to have led to the 1:1 ratio of the oral cavity relative to the pharyngeal cavity that is considered to be a crucial prerequisite for the evolution of complex speech articulation^{1,59}.”

It is worth noting that we would have missed this very interesting point if we had exclusively focussed on vocal sex dimorphism, rather than departures from the acoustic allometry in males.

2. The authors used sexual size dimorphism "as an indicator of the intensity of sexual selection pressures acting on male body size in a given species". This a reasonable decision for measuring sexual selection pressures acting specifically *on male body size*, but one cannot extrapolate from sexual selection on body size to the intensity of sexual selection more broadly (Plavcan, 2004; 2011; 2012), as the authors do, e.g., "When investigating the effect of sexual selection, we found that sexual size dimorphism did not predict F0 across taxa..."

The primary objective of this study was to investigate whether sexual selection pressures for larger males lead to acoustic size exaggeration in male vocal signals, and whether post-copulatory sexual selection pressures for increased sperm production lead to a relaxation in acoustic size exaggeration, rather than on sexual selection pressures related to the intensity of mating competition.

To clarify this we now provide explicit headings in the results section that make our objectives clear. In addition, we have changed:

“When investigating the effect of sexual selection, we found that...”

to

“When investigating the effect of sexual selection for large male body size, we found that ...”

and

“This indicates that sexual selection is likely to be a key force...”

To

“This indicates that sexual selection for increased male body size is likely to be a key force...”

There are two main reasons for this. First, size dimorphism is only a modest indicator of one form mating competition: overt aggression. The

frequency and intensity of male-male agonism explained only 48% of the variation in sexual size dimorphism in a sample of 128 anthropoid primate species (Plavcan, 2012).

We agree with this, but as stated above, we are not interested in modelling mating competition in general, but specifically the effect sexual selection for male size on acoustic exaggeration. It is also worth noting that size dimorphism could arise due to inter-sexual selection pressures for larger males as mating partners (e.g. Charlton et al 2007, 2012). Thus, our measure of sexual selection for increased male size does not solely reflect intra-sexual selection pressures/male-male competition.

Second, mating competition takes many forms other than fighting. Because there are numerous forms of mating competition (e.g., sperm competition, scramble competition) that do not produce large size dimorphism, sexual size dimorphism is a poor proxy for the intensity of mating competition overall (e.g., Plavcan, 2012). Thus, another measure of the intensity of sexual selection should be used, such as breeding system.

While we agree that mating system could constitute a useful index of overall sexual selection intensity, in order to use such an index we would need access to a reasonable sample of species for each of the different types of mating system. Our data set (now containing 72 species across 9 mammalian orders) has 50 polygynous species, but only 8 monogamous species, 3 polyandrous species, and 8 promiscuous species. The mating system of three species (humans, brown lemurs and black and white ruffed lemurs) is not well defined and therefore classed as “variable” (Supplementary Table S1). Because only 9% of mammals are monogamous and the vast majority of non-primates are polygynous, our dataset is representative of the distribution of mating systems across mammals.

In response to this comment, we now control for more general sexual selection pressures by including mating (breeding) system as a covariate in all of the global PGLS regression models used to test our hypotheses. In each case, the best supported model did not include mating system as a covariate (i.e. after model selection based on the lowest AICc value), nor was mating system a statistically significant covariate in any of the models (see supplementary tables S2-S7).

3. A recent paper by Puts and colleagues (2016) is highly relevant to the present work and should be discussed in the context of the present study.

We now discuss the findings of Puts et al on lines 184-187 of the manuscript:

“Indeed, recent findings in anthropoid primates show how sexual dimorphism in F0 increases during evolutionary transitions towards polygyny, and decreases during transitions towards monogamy³³, further emphasizing that F0 is a sexually selected component of mammal vocalisations.”

These authors explored the evolution of F0 sexual dimorphism in relation to breeding system, habitat, and body size dimorphism in anthropoid primates. Of particular relevance is their finding relating sexual F0 dimorphism to breeding system and body size dimorphism: Low male (relative to female) vocalisations related negatively to size dimorphism, once breeding system was controlled.

Please note that increased male over female size dimorphism results in ratios > 1 and lower male versus female F0 or formants will result in a ratio < 1 ; hence, a negative relationship is expected when ratios of size dimorphism and vocal dimorphism are correlated because larger males (or females) would be expected to produce lower call frequencies (F0 and formants) due to their larger vocal anatomy. This relationship is shown very clearly in Fig. 1 of the current study (depicting the acoustic allometry), and also by our PGLS regressions, in which male size is negatively correlated to F0 and formants.

They hypothesized that this is because the function of low-frequency male vocalizations is precisely for *avoiding* fights (via size exaggeration).

Because Puts *et al* did not consider formants in their analysis of anthropoid primates, we assume this hypothesis is based on the fact that humans perceive low F0 as equating to large body size. In actual fact, F0 is an unreliable cue to size in humans and most other mammals (explaining less than 2% variance in adult human size, for example) because laryngeal growth is relatively unrestrained and strongly influenced by androgens. Hence, F0 is unlikely to be maintained as a cue to body size due to selection pressures on receivers to discriminate between reliable and unreliable signals.

In contrast, formant spacing is often a reliable cue to mammalian body size because it reflects skull size, which is tightly linked to overall adult body size in most mammals. This provides a better foundation for the evolution of adaptations that allow species to exaggerate body size via lowering formant frequencies (see Taylor, Charlton & Reby 2016 for an extended discussion). Our results strongly support this contention.

Thus, on the one hand, inter- or intrasexual mating competition may produce both acoustic and body size dimorphisms. Meanwhile, on the other hand, because large body size helps males win fights, and low vocalisations may help males avoid fights, the two may be inversely related when the overall intensity of male mating competition is controlled.

This is precisely what we find in our analysis: larger relative male body size = lower F0 and ΔF – an inverse (negative) relationship.

We now control for the overall intensity of male competition by including mating system as a covariate in our global PGLS regression models (also see previous responses).

These results are also similar in principle to those of Dunn et al. (2015; cited in the present work), who found that male howler monkeys do not evolve acoustic anatomical dimorphisms when their mating competition predominantly takes a form other than direct fighting (sperm competition, in the case of howlers). Indeed, in the present work, relatively low frequency (in F0 and delta F) male calls occurred in species with relatively small testes, suggesting that the form of mating competition rather than its overall intensity influences the evolution of male calls, and cautioning against using a (somewhat weak) measure of male agonism as a proxy for overall mating competition.

We now include mating system (monogamous, polygynous, polyandrous, promiscuous, or variable) in our analysis, and by doing this we control for other more general sexual selection pressures aside from those specifically contingent on male body size.

We would however like to emphasise that we do not use male sexual size dimorphism as a proxy for mating competition, only as a proxy for sexual selection pressures on male body size (which should also reflect selection pressures to sound larger). We have now made this much clearer to the reader, and apologise for not having made this point sufficiently clear in our original manuscript.

4. The authors collected acoustic data only from vocalisations utilized in presumptive "sexual" calls. However, it should be noted that the functions of calls are not known with certainty, and other "non-sexual" calls may in fact be relevant to mating competition. Moreover, given that sexual selection pressures produce low frequency vocalisations by shaping the underlying anatomy (larynges, vocal sacs, etc.), the effects of sexual selection should be observable across the range of vocalisation types produced in a species.

This assumption fails to account for the behavioural component of call production. Indeed, sexual calls often involve specific gestures. For example, in red and fallow deer, males retract their larynges to lower formant frequencies and sound as large as possible during call production. Again, this is not observed in other call-types and does not affect their frequency components.

Moreover, the production of sexual calls can also involve specific anatomical specialisations that are not recruited for the production of nonsexual calls. For instance, male koalas use a non-laryngeal source to produce the extremely low F0 of bellow vocalizations (Charlton et al 2013, Curr. Biol.). Other koala vocal signals have a much higher F0, presumably because they are produced by vocal fold vibration in the larynx.

In response to the referee's comment, we now also include data from nonsexual calls and enter call-type (sexual or nonsexual) as a covariate in the analysis to control for differences in acoustic structure generated by

adaptations of vocal anatomy (i.e. descended and mobile larynges, and non-laryngeal sound sources) that are only used to produce sexual calls.

Finally, in order to measure sexual dimorphism in vocal frequencies (see above), it will be necessary to measure "non-sexual" calls, as females do not produce "sexual" calls in some species.

While the primary objective of this study was to investigate if departures from the acoustic allometry (rather than sexual dimorphism in vocal frequencies) were linked to sexual size dimorphism we agree that this approach has some merit and now include acoustic data derived from nonsexual calls (see previous response).

REVIEWERS' COMMENTS:

Reviewer #1 (Remarks to the Author):

I think the authors have done an exemplary job addressing the reviewers' concerns and that the paper is substantially improved because of their careful revisions. Importantly, I believe that their new PGLS analyses are appropriate, the data set is larger, and the results are robust.

Dan Blumstein

Reviewer #2 (Remarks to the Author):

The authors have addressed my previous comments and revised the ms appropriately.

Reviewer #3 (Remarks to the Author):

I appreciate the authors' attention to my comments and suggestions, particularly their clarifications regarding the focus of their work, as well as their inclusion of new data (e.g., mating system and non-sexual calls) in their analyses. I have one residual overarching comment concerning the authors' predictions and interpretation of their results. Essentially, it is not clear that the present data can be used to directly test whether FO or delta-F function as size exaggerators.

I am referring specifically to the following statements: L57 "...whether sexual selection pressures on male body size drive the evolution of putative acoustic size exaggeration across a wider range of mammalian taxa remains to be investigated." L71 "We...predict that males from mating systems with strong selection pressures for large male body size, as evidenced by greater male sexual size

dimorphism, will produce lower call frequencies than expected for their size." L167 "When investigating the effect of sexual selection for large male body size we found that sexual size dimorphism did not predict F0 across taxa. The lack of a relationship between sexual size dimorphism and F0 is not surprising as F0 is generally a poor predictor of adult male body mass within species⁵. Our results, therefore, support the hypothesis that sexual selection does not favour the use of F0 as an acoustic size exaggerator."

The authors' logic is that if it is important for a male to be large, then he will also benefit from exaggerating his apparent size acoustically, and that if it is less beneficial for a male to be large, then he will benefit less from exaggerating his size acoustically. This prediction makes intuitive sense and is supported with regard to ΔF but not with regard to F0. One could conclude from these results that (probably sexual) selection favoring relatively large male size does not tend to favour the use of F0 as an acoustic size exaggerator. However, one *cannot* conclude from the results that (a) F0 is not an acoustic size exaggerator, or that (b) "sexual selection does not favour the use of F0 as an acoustic size exaggerator." These may seem like a subtle distinctions, but I think that they are important ones. My reasoning is as follows:

1. A function of F0 in exaggerating size was not measured in any species in the present research. Given the cross-species (and to a lesser degree, within-species) relationships between size and low F0, it is reasonable to suppose that a low F0 sounds larger to other mammals, as we know that it does in humans.

2. The selection pressures that produce size dimorphism are only one set of potential influences on the importance of acoustically exaggerating size across species. An animal does not have to be large to benefit from exaggerating its size, and there may be contexts in which selection does not favour relatively large male size, but in which there is still sexual selection to exaggerate male size acoustically. There may also be contexts in which selection *does* favour relatively large body size but it nevertheless does not favour size exaggeration via F0.

In this vein, the authors replied to one of my previous comments: "...F0 is an unreliable cue to size in humans and most other mammals (explaining less than 2% variance in adult human size, for example)... Hence, F0 is unlikely to be maintained as a cue to body size due to selection pressures on receivers to discriminate between reliable and unreliable signals. In contrast, formant spacing is often a reliable cue to mammalian body size..."

First, I would point out that formant spacing is not much better than F0 as a cue to body size; delta-F explains <5% of the variance in men's body mass (Pisanski et al., 2014, Anim Behav 95: 89). Second, a function of F0 or delta-F as a size exaggerator does not depend on either acoustic trait being a reliable cue to size. It is necessary only that the acoustic trait affects perceptions of size. (See, e.g., Rendall, Vokey, & Nemeth, 2007 for possible reasons why these perceptions are maintained despite the modest size-acoustic associations within species.) For example, eyespots on butterflies are not reliable cues to the insects' size, but they nevertheless function in increasing the appearance of size to would-be predators.

Thus, we can't say from the present results that "sexual selection does not favour the use of F0 as an acoustic size exaggerator." Rather, we must say that (probably sexual) selection for large male size does not systematically favour low F0 as a size exaggerator. Sexual selection may still favour low F0 as a size exaggerator (as seems to have been the case in humans, and as suggested by the present results regarding sperm competition vs. precopulatory competition); it just may do so even in species that are not highly dimorphic in body size, and it may *not* do so even in some species that *are* highly dimorphic in body size. This is an important distinction because it bears on the cross-species function of low pitch. Previous authors and data have suggested that this function is to exaggerate size, and present data do not strongly challenge this. Rather, the present data provide new evidence about the conditions under which putative acoustic size exaggerations evolve.

Minor Editing

L22 "in mating systems with sexual selection pressures for large male body size, confirming that sexual selection favours the use of ΔF as an acoustic size exaggerator."

To be more accurate, this should be changed to something like "in species with selection for larger male than female body size, suggesting that sexual selection favours the use of ΔF as an acoustic size exaggerator."

L25 "In contrast, there was no relationship between F0 and male size dimorphism."

This should read "body size dimorphism" as males are presumably not dimorphic-the sexes are.

Reference

Rendall, D., Vokey, J. R., & Nemeth, C. (2007). Lifting the curtain on the Wizard of Oz: biased voice-based impressions of speaker size. *Journal of Experimental Psychology Human Perception and Performance*, 33(5), 1208-1219. doi: 2007-14662-016 [pii]10.1037/0096-1523.33.5.1208

Please find our response to the reviewer's comments (in bold red) below. Our responses follow the review comments in normal black font.

(Remarks to the Author):

I appreciate the authors' attention to my comments and suggestions, particularly their clarifications regarding the focus of their work, as well as their inclusion of new data (e.g., mating system and non-sexual calls) in their analyses. I have one residual overarching comment concerning the authors' predictions and interpretation of their results. Essentially, it is not clear that the present data can be used to directly test whether F0 or delta-F function as size exaggerators.

We agree with this statement. Playback experiments would be needed to directly test whether F0 or formant spacing function as size exaggerators in a given system. We hope that our comparative investigation inspires this type of future research.

I am referring specifically to the following statements: L57 "...whether sexual selection pressures on male body size drive the evolution of putative acoustic size exaggeration across a wider range of mammalian taxa remains to be investigated." L71 "We...predict that males from mating systems with strong selection pressures for large male body size, as evidenced by greater male sexual size dimorphism, will produce lower call frequencies than expected for their size." L167 "When investigating the effect of sexual selection for large male body size we found that sexual size dimorphism did not predict F0 across taxa. The lack of a relationship between sexual size dimorphism and F0 is not surprising as F0 is generally a poor predictor of adult male body mass within species⁵. Our results, therefore, support the hypothesis that sexual selection does not favour the use of F0 as an acoustic size exaggerator."

The authors' logic is that if it is important for a male to be large, then he will also benefit from exaggerating his apparent size acoustically, and that if it is less beneficial for a male to be large, then he will benefit less from exaggerating his size acoustically. This prediction makes intuitive sense and is supported with regard to delta-F but not with regard to F0.

We agree. This is our rationale and our findings support the contention that

formant spacing is used as an acoustic size exaggerator across mammals.

One could conclude from these results that (probably sexual) selection favoring relatively large male size does not tend to favour the use of F0 as an acoustic size exaggerator. However, one *cannot* conclude from the results that (a) F0 is not an acoustic size exaggerator, or that (b) "sexual selection does not favour the use of F0 as an acoustic size exaggerator." These may seem like a subtle distinctions, but I think that they are important ones. My reasoning is as follows:

1. A function of F0 in exaggerating size was not measured in any species in the present research. Given the cross-species (and to a lesser degree, within-species) relationships between size and low F0, it is reasonable to suppose that a low F0 sounds larger to other mammals, as we know that it does in humans.

2. The selection pressures that produce size dimorphism are only one set of potential influences on the importance of acoustically exaggerating size across species. An animal does not have to be large to benefit from exaggerating its size, and there may be contexts in which selection does not favour relatively large male size, but in which there is still sexual selection to exaggerate male size acoustically. There may also be contexts in which selection *does* favour relatively large body size but it nevertheless does not favour size exaggeration via F0.

In this vein, the authors replied to one of my previous comments: "...F0 is an unreliable cue to size in humans and most other mammals (explaining less than 2% variance in adult human size, for example)... Hence, F0 is unlikely to be maintained as a cue to body size due to selection pressures on receivers to discriminate between reliable and unreliable signals. In contrast, formant spacing is often a reliable cue to mammalian body size..."

First, I would point out that formant spacing is not much better than F0 as a cue to body size; delta-F explains <5% of the variance in men's body mass (Pisanski et al., 2014, *Anim Behav* 95: 89).

The Pisanski et al paper 2014 actually states the following:

"The analysis revealed only weak or marginal negative relationships between F0 and height or weight within either sex. In fact, F0 accounted for less than 2% of the variance in height or weight within sexes, whereas individual formant-based VTL estimates could explain upwards of 10% of the variance in height or weight within sexes."

Hence, formant spacing does explain more than 2 % variation in adult human body size (as stated above: "individual formant-based VTL estimates could explain upwards of 10% of the variance in height or weight within sexes"). The reviewer also cites body weight, which, in humans is a relatively poor proxy of

skeletal size (compared to height, for example) due to body fat variation associated with western diets.

Second, a function of F0 or delta-F as a size exaggerator does not depend on either acoustic trait being a reliable cue to size. It is necessary only that the acoustic trait affects perceptions of size. (See, e.g., Rendall, Vokey, & Nemeth, 2007 for possible reasons why these perceptions are maintained despite the modest size-acoustic associations within species.) For example, eyespots on butterflies are not reliable cues to the insects' size, but they nevertheless function in increasing the appearance of size to would-be predators.

Thus, we can't say from the present results that "sexual selection does not favour the use of F0 as an acoustic size exaggerator." Rather, we must say that (probably sexual) selection for large male size does not systematically favour low F0 as a size exaggerator.

Because we accept that that F0 cannot be ruled out as a size exaggerator in some species we have now rephrased lines 259-261:

“Our results, therefore, support the hypothesis that sexual selection does not favour the use of F0 as an acoustic size exaggerator”

“Our results, therefore, support the hypothesis that sexual selection for large male size does not systematically favour the use of F0 as an acoustic size exaggerator”

Sexual selection may still favour low F0 as a size exaggerator (as seems to have been the case in humans, and as suggested by the present results regarding sperm competition vs. precopulatory competition); it just may do so even in species that are not highly dimorphic in body size, and it may *not* do so even in some species that *are* highly dimorphic in body size. This is an important distinction because it bears on the cross-species function of low pitch. Previous authors and data have suggested that this function is to exaggerate size, and present data do not strongly challenge this. Rather, the present data provide new evidence about the conditions under which putative acoustic size exaggerations evolve.

We are aware that, in humans, biases mean that individuals with lower F0s are typically perceived as larger even though F0 explains a very small proportion of adult intra-sex body size variation, possibly as a consequence of over-generalisation (Rendall et al 2007). However, because no such phenomenon has been demonstrated in non-human mammals, it is speculative to assume that lowering F0 equates to exaggerating body size in non-human mammals. Since the current study incorporates 71 nonhuman mammal species, we feel it would be inappropriate to generalize findings in humans to all mammals.

Minor Editing

L22 "in mating systems with sexual selection pressures for large male body size, confirming that sexual selection favours the use of ΔF as an acoustic size exaggerator."

To be more accurate, this should be changed to something like "in species with selection for larger male than female body size, suggesting that sexual selection favours the use of ΔF as an acoustic size exaggerator."

This has now been changed.

L25 "In contrast, there was no relationship between F0 and male size dimorphism."

This should read "body size dimorphism" as males are presumably not dimorphic-the sexes are.

This has now been changed.

Reference

Rendall, D., Vokey, J. R., & Nemeth, C. (2007). Lifting the curtain on the Wizard of Oz: biased voice-based impressions of speaker size. *Journal of Experimental Psychology Human Perception and Performance*, 33(5), 1208-1219. doi: 2007-14662-016 [pii]10.1037/0096-1523.33.5.1208